# An Extensible Benchmark Suite for Learning to Simulate Physical Systems

**Karl Otness**
Courant Institute of Mathematical Sciences
New York University
`karl.otness@nyu.edu`

**Arvi Gjoka**
Courant Institute of Mathematical Sciences
New York University
`arvi.gjoka@nyu.edu`

**Joan Bruna**
Courant Institute of Mathematical Sciences
New York University

**Daniele Panozzo**
Courant Institute of Mathematical Sciences
New York University

**Benjamin Peherstorfer**
Courant Institute of Mathematical Sciences
New York University

**Teseo Schneider**
University of Victoria

**Denis Zorin**
Courant Institute of Mathematical Sciences
New York University

## Abstract

Simulating physical systems is a core component of scientific computing, encompassing a wide range of physical domains and applications. Recently, there has been a surge in data-driven methods to complement traditional numerical simulation methods, motivated by the opportunity to reduce computational costs and/or learn new physical models leveraging access to large collections of data. However, the diversity of problem settings and applications has led to a plethora of approaches, each one evaluated on a different setup and with different evaluation metrics. We introduce a set of benchmark problems to take a step towards unified benchmarks and evaluation protocols. We propose four representative physical systems, as well as a collection of both widely used classical time integrators and representative data-driven methods (kernel-based, MLP, CNN, nearest neighbors). Our framework allows evaluating objectively and systematically the stability, accuracy, and computational efficiency of data-driven methods. Additionally, it is configurable to permit adjustments for accommodating other learning tasks and for establishing a foundation for future developments in machine learning for scientific computing.

## 1   Introduction

Computational modeling of physical systems is a core task of scientific computing. Standard methods rely on discretizations of explicit models typically given in the form of partial differential equations (PDEs). Machine learning techniques can extend these techniques in a number of ways. In some cases, a closed system of analytic equations relating all variables may not be available (e.g., a constitutive relation for a material may not be known). In other cases, while a full analytic description of a system is available, a traditional solution may be too costly (e.g., turbulence) or can be sped up substantially using data-driven reduced-order models. However, despite promising results, a

35th Conference on Neural Information Processing Systems (NeurIPS 2021) Track on Datasets and Benchmarks.

successful adoption of these data-driven approaches into scientific computing pipelines requires a solid and exhaustive assessment of their performance—a challenging task given the diversity of physical systems, corresponding data-driven approaches, and the lack of standardized sets of problems, comparison protocols, and metrics.

We focus on the setting where the physical model is unavailable during training, mimicking situations in computational science and engineering with ample data and a lack of models. One can generally distinguish two different flavors of physical simulation with different associated computational cost: those that map a high-dimensional state space into another high-dimensional space (as in temporal integration schemes, mapping the state of the system at one time step to the next), or from a high-dimensional input space to a lower-dimensional output (as in surrogate models, mapping the initial conditions to a functional of the solution). While this distinction also applies to data-driven approaches, another critical aspect emerges from the choice of input data distribution. We identify two extremes: the *narrow* data regime, where initial conditions are sampled from a low-dimensional manifold (even within a high-dimensional state space), and the *wide* regime, where initial conditions span a truly high-dimensional space. As could be expected, narrow data regimes define an easier prediction task where data-driven methods can potentially 'bypass the physics,' whereas wide regimes require models with enough encoded physical priors in order to beat the curse of dimensionality. Therefore, such choice of data distribution is a critical component of any data-driven physical simulation benchmark.

In this work, we introduce an extensible benchmark suite, including: **(1)** an extensible set of simple, yet representative, physical models with a range of training and evaluation (test) setups, as well as reference, high-accuracy numerical solutions to benchmark data-driven methods, **(2)** reference implementations of traditional time integration schemes, which are used as baselines for evaluation, and **(3)** implementations of widely used data-driven methods, including physics-agnostic multi-layer perceptrons (MLPs), convolutional neural networks (CNNs), kernel machines and non-parametric nearest neighbors. Our benchmark suite is modular, permitting extensions with limited code changes, and captures both 'narrow' and 'wide' regimes by appropriately parametrizing the set of initial conditions.

Our analysis reveals two important conclusions. First, even in the simplest physical models, current data-driven pipelines, while providing qualitatively acceptable solutions, are quantitatively far from directly numerically integrating physical models, and this performance gap appears unfeasible to close by merely scaling up the models and/or the dataset size. In other words, the cost of ignoring the physics is high, even for the simplest physics, and cannot in general be compensated by data, matching insights that have been obtained in other scientific computing settings [5, 55]. Next, and more importantly, our simple $L^2$-based nearest neighbor regressor is used to calibrate how 'narrow' the learning task is. Our finding is that even for seemingly complex systems, such as the incompressible Navier-Stokes systems, such a naive predictor outperforms most deep-learning-based models in the narrow regime—thus providing a simple calibration of the true difficulty of the simulation task, that we advocate should be present in every future evaluation.

## 2 Related work

Machine learning is used in physical simulation in a number of interrelated ways. Some important uses include reduced-order/surrogate modeling, learning constitutive models or more generally compact analytic representations from data. A unifying theme of these applications of machine learning is automatic construction of parametric models capable of reproducing the behavior of physical systems for a sufficiently broad range of initial data, boundary conditions and other system parameters. The purpose of these representations varies from acceleration (e.g., surrogate machine learning models are used to accelerate optimization), to automatic construction of multiscale models (learning macroscopic constitutive laws from microscopic simulation), to inferring compact descriptions of unknown representations from experimental data.

The purpose of our proposed benchmarks is to enable comparisons of different learning-based methods in terms of their accuracy and efficiency. We briefly review two streams of learning methods for physical systems. **(1)** One line of work aims to understand how neural networks can be structured and trained to reproduce known physical system behavior, with the goal of designing general methods applicable in a variety of settings [8, 42, 4, 41, 37, 38, 26, 10, 50, 51]. Our

benchmark cases fit primarily into this category. **(2)** Another line of research aims to develop a variety of techniques to accelerate solving PDEs. Typically, these methods are developed for specific PDEs and a specific restricted range of problems: for example, fluid dynamics problems [39, 16, 56], with particular applications to cardiovascular modeling [25, 19] and aerodynamics [53]; or solid mechanics simulation tasks, including stresses [29, 24, 27, 15, 22, 23]. In cases where the governing equations are not given, the learning task becomes approximating them from data [30, 7, 1, 9, 2, 28, 3, 43, 44, 46, 45, 52, 35].

## 3 Background and problem setup

**PDEs, dynamical systems, and time integration**   Consider a PDE of the form $\partial_t u = \mathcal{L}(u)$, where $u$ is the unknown function and $\mathcal{L}$ is a possibly nonlinear operator that includes spatial derivatives of $u$. By discretizing in space, one obtains a dynamical system

$$\dot{x}(t) = f(x(t)) \tag{1}$$

with an $N$-dimensional state $x(t) \in \mathbb{R}^N$ at time $t \in [0, T]$. The function $f$ is assumed to be Lipschitz to ensure solution uniqueness and the initial condition is denoted as $x_0 \in \mathbb{R}^N$. A PDE of a higher order in time can be reduced to the first-order form in the standard way, e.g., if we have a second-order system $\ddot{q}(t) = f(q(t))$, then we consider its formulation via position $q$ and momentum $p$ as a first-order system with $x = [q; p]$: $[\dot{q}(t); \dot{p}(t)] = [p(t); f(q(t))]$. To numerically integrate (1), we choose time steps $0 = t_0 < t_1 < \cdots < t_K = T$. Then, a time integration scheme (e.g., [49, 12, 11]) gives an approximation $x_k \approx x(t_k)$ of the state $x(t_k)$ at each time step $k = 1, \ldots, K$. A list of the schemes we use along with details is given in Appendix A.

**Problem setup and learning problems**   Given $M$ initial conditions $x_0^{(1)}, \ldots, x_0^{(M)} \in \mathbb{R}^N$ and the corresponding $M$ trajectories $X^{(i)} = [x_0^{(i)}, \ldots, x_K^{(i)}] \in \mathbb{R}^{N \times (K+1)}, i = 1, \ldots, M$ obtained with a time integration scheme from dynamical system (1), we consider the following two learning problems, both of which aim to learn the physical model of the problem, viewed as unknown, from trajectory samples: **(1)** Learning an approximation $\tilde{f}$ of the right-hand side function $f$ in Eq. (1). This gives an approximate $\dot{\tilde{x}}(t) = \tilde{f}(\tilde{x}(t))$ that is then numerically integrated to produce a trajectory $\tilde{X}$ for an initial condition $\tilde{x}_0$. The aim is that $\tilde{X}$ approximates well the true trajectory $X$ obtained with $f$ from (1) for the same initial condition. **(2)** Directly learning the next steps in the trajectory from the current one, i.e. predict $x_k^{(i)}$ given $x_{k-1}^{(i)}$.

To assess the learned models, we evaluate them on their ability to produce good approximate trajectories from randomly sampled initial conditions, by either integration or direct step prediction. During evaluation, we use initial conditions drawn independently from those used to produce training data, both from the same distribution as the training samples, as well as from a distribution with support outside the training range. We train networks on data sets of various sizes. For details, see Appendix B.

## 4 Benchmark systems

We consider four physical systems, illustrated in Figure 1: a single oscillating spring, a one-dimensional linear wave equation, a Navier-Stokes flow problem, and a mesh of damped springs. These systems represent a progression of complexity: the spring system is a linear system with a low-dimensional space of initial conditions and low-dimensional state; the wave equation is a low-dimensional linear system with a (relatively) high-dimensional state space after discretization; the Navier-Stokes equations are nonlinear and we consider a setup with low-dimensional initial conditions and a high-dimensional state space; finally, the spring mesh system has both high-dimensional initial conditions as well as high-dimensional states. Additionally, the proposed spring system and Navier-Stokes problems represent diffusion-dominated and advection-dominated (for sufficiently low viscosity) PDE behaviors, as well as variability in initial conditions with fixed domain (spring system) and variable domain (Navier-Stokes). These varying complexities provide an opportunity to test methods on simpler systems and the ability to examine changing performance as system size increases, both in terms of the state dimension, and the initial condition distribution. The ground truth models for the spring, wave, and spring mesh systems with classical time integrators are implemented

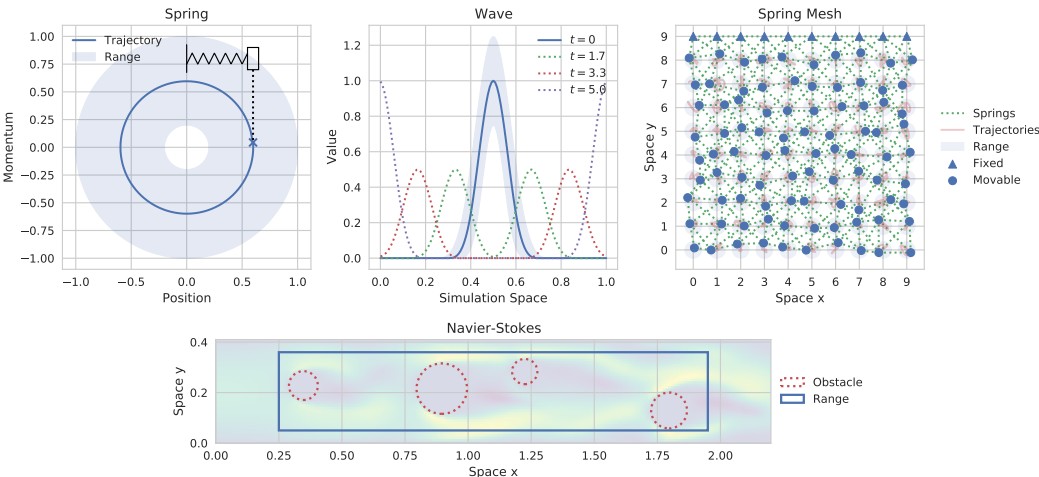

Figure 1: Representative visualizations of the four systems, depicting the results and ranges of initial condition sampling. Each has two state components: for the Navier-Stokes system, a flow velocity and a pressure field, and for the other three a position $q$ and momentum $p$.

using NumPy [13], SciPy [54], and accelerated, where possible, with Numba [21]. The Navier-Stokes snapshots are generated using PolyFEM [47], a finite element library.

These systems were chosen in an effort to reflect the variety of systems used for testing in this area, while unifying choices of particular formulations. Past works have chosen systems of the types featured here: simple oscillators (both spring and pendulum [8]), particle systems with various interaction laws (gravity, spring forces, charges, cloth simulations, etc. [6, 18, 42, 4, 34]), and fluid-flow systems (with various sorts of obstacles, airfoils or cylinders [51, 34]). We make particular selections here in an effort to unify systems of interest and facilitate comparisons across experiments by providing a shared set of tasks which can be used for development and testing of machine learning methods.

Some examples of initial condition selection for each system are illustrated in Figure 1. The ground truth for the spring, wave, and spring mesh systems consists of the state variables $(q, p)$ for position and momentum, and their associated derivatives $(\dot{q}, \dot{p})$. For the Navier-Stokes system the state consists of flow velocities, and a pressure field, along with approximated time derivatives for each.

Table 1 lists the parameters used to generate trajectories for training and evaluation. Training sets of three sizes are generated, each containing the specified number of trajectories. The systems are integrated at the listed time step sizes, but the ground truth data is subsampled further by the factor shown after ÷ in the table: the integration schemes are run at a smaller time step and intermediate computations are discarded. Each larger training set is a strict superset of its predecessor to ensure that previous training samples are never removed.

## 4.1 Spring

We simulate a simple one-dimensional oscillating spring. In this system, the spring has zero rest length, and both the oscillating mass and spring constant are set to 1. The spring then exerts a force inversely proportional to the position of the mass $q$: $\dot{p}(t) = -q$ and $\dot{q}(t) = p$.

The energy of the system is proportional to $r = q^2 + p^2$ which is the radius of the circle in phase space. To sample initial conditions, we first sample a radius uniformly, then choose an angle theta uniformly. This produces a uniform distribution over spring system energy levels and starts at an arbitrary point in the cycle. The spring system has a closed-form solution: $(q(t), p(t)) = (r \sin(t + \theta_0), r \cos(t + \theta_0))$ where $r$ is the radius of the circle traced in phase space (the energy of the spring) and $\theta_0$ is the phase space angle at which the oscillation will start. While this closed form solution is useful, for consistency with our other systems, we generate snapshots of the spring system by numerical integration. Simulations of the spring system always run through one period. For "in-distribution"

training values, the radius is selected in the range $(0.2, 1)$ and "out-of-distribution" radii are chosen from $(1, 1.2)$.

## 4.2 Wave

This benchmark system is similar to the one used in Peng and Mohseni [33]. Consider the wave equation with speed $c = 0.1$

$$\partial_{tt} u = c^2 \partial_{xx} u \,, \tag{2}$$

on a one-dimensional spatial domain $[0, 1)$ with periodic boundary conditions. We represent this second-order system as a first-order system and discretize in space to obtain

$$\begin{bmatrix} \dot{q}(t) \\ \dot{p}(t) \end{bmatrix} = \begin{bmatrix} 0 & I \\ c^2 D_{xx} & 0 \end{bmatrix} \begin{bmatrix} q(t) \\ p(t) \end{bmatrix} \,, \tag{3}$$

where $D_{xx} \in \mathbb{R}^{n \times n}$ corresponds to the three-point central difference approximation of the spatial derivative $\partial_{xx}$ and the matrices $I$ and $0$ are the identity and zero matrix, respectively, of appropriate size. We discretize in space with $n = 125$ evenly spaced grid points and evolve the system following the dynamics described above.

Initial conditions are sampled with an initial pulse in the $q$ component centered at $0.5$. All initial conditions have zero momentum. The initial pulse is produced by a spline kernel as described in [33]:

$$s(x) = \frac{10}{p_w} \cdot |x - 0.5| \,, \qquad h(s) = p_h \cdot \begin{cases} 1 - \frac{3}{2}s^2 + \frac{3}{4}s^3 & \text{if } 0 \leq s \leq 1 \\ \frac{1}{4}(2 - s)^3 & \text{if } 1 < s \leq 2 \\ 0 & \text{else} \end{cases}$$

where the width and height of the pulse are scaled by parameters $p_w$ and $p_h$, respectively. The spline kernel pulse is then $h(s(x))$ for $x \in [0, 1)$, evaluated at the discretized grid points.

For "in-distribution" samples, parameters $p_w, p_h$ are both chosen uniformly in the range $(0.75, 1.25)$ and "out-of-distribution" runs sample uniformly from $(0.5, 0.75) \cup (1.25, 1.5)$. All trajectories are integrated until $t = 5$ when the wave has traveled through half a period.

## 4.3 Spring mesh

This system manipulates a square grid of particles connected by springs, in a two dimensional space, and can be considered a simplified version of deformable surface and volume systems (cf. [34]). The particles all have mass 1, and are arranged into a unit grid. Springs are added along the axis-aligned edges and diagonally across each grid square, with rest lengths selected so that the regularly-spaced particles are in a rest position.

In this work we use a $10 \times 10$ grid where the top row of particles is fixed in place. Initial conditions are sampled by choosing a perturbation for the position of each non-fixed spring. These perturbations are chosen as uniform vectors inside a circle with radius 0.35. Out-of-distribution perturbations are chosen uniformly in a ring with inner radius 0.35 and outer radius 0.45. The sampled initial conditions all have zero momentum.

In this system, a spring between particles $a$ and $b$ exerts a force:

$$F_{ab} = -k \cdot \left( \|q_a - q_b\|_2 - \ell_{ab} \right) \frac{q_a - q_b}{\|q_a - q_b\|_2} - \gamma \dot{q}_a \tag{4}$$

where $\ell_{ab}$ is the rest length of the spring, $\gamma = 0.1$ is a parameter controlling the magnitude of an underdamped velocity-based decay, and $k = 1$ is the spring constant.

## 4.4 Navier-Stokes

We consider the standard Navier-Stokes equation over a domain $\Omega$ (cf. [34, 51])

$$\left. \begin{array}{l} \rho \dfrac{\partial u}{\partial t} + \rho(u \cdot \nabla)u - \nu \Delta u + \nabla p = b \\ \nabla \cdot u = 0 \\ u(0) = u_0 \end{array} \right\} \text{ on } \Omega \times (0, T) \,, \quad \left. \begin{array}{l} u = d \\ \nu \dfrac{\partial u}{\partial n} + pn = g \end{array} \right\} \text{ on } \partial \Omega_D \times (0, T) \,,$$

Table 1: Dataset sizes and simulation parameters

| System | # Train Trajectories | # Eval Trajectories | Time Step Size | | # Steps |
|---|---|---|---|---|---|
| Spring | 10, 500, 1000 | 30 | 0.00781 | $\div 128$ | 805 |
| Wave | 10, 25, 50 | 6 | 0.00049 | $\div 8$ | 10204 |
| Spring Mesh | 25, 50, 100 | 15 | 0.00781 | $\div 128$ | 805 |
| Navier-Stokes | 25, 50, 100 | 5 | 0.08 | $\div 1$ | 65 |

where $u \colon \Omega \times (0, T) \to R^2$ is the velocity at time $t \in (0, T)$ of a fluid with kinematic viscosity $\nu$ and density $\rho$, $p \colon \Omega \times (0, T) \to R$ is the pressure and $\partial \Omega_D$ and $\partial \Omega_N$ are the Dirichlet and Neumann boundary conditions, respectively. In our setup we use the finite element method (FEM) to solve the PDE using mixed discretization: quadratic polynomial for the velocity and linear for pressure. In our experiment the domain $\Omega$ is a rectangle $0.22 \times 0.41$ with a randomly generated set of circular obstacles. We start with $u_0 = 0$ and specify a velocity on the left boundary of $u(0, y) = (6(1 - e^{-5t})(0.41 - y)y/0.1681, 0)$, zero on the top and bottom, and zero Neumann on the right side ($g = 0$). We solve the system using PolyFEM [47] using $dt = 0.08$ and backward differentiation formula (BDF) of order 3 for the time integration.

We sample obstacles into two configurations: a single obstacle, or a set of four. In each case, we sample the obstacles leaving a margin of 0.05 between each circle, and a margin of 0.25 from the left and right sides, and 0.05 between the top and bottom. Otherwise, each obstacle is determined by first sampling a radius, then sampling a center from the valid space, respecting the margins. If the sampled obstacle is too close to an existing circle, it is discarded and a new sample is drawn. In-distribution obstacles have radii in the range $(0.05, 0.1)$ and out-of-distribution radii are drawn from $(0.025, 0.05)$.

## 5   Numerical experiments

**Experimental setup**   We apply several basic learning methods to the datasets developed in this work: $k$-nearest neighbor regressors, a neural network kernel method, several sizes of feed-forward MLPs, and a variety of CNNs. Details of the architectures and the training protocol are provided in supplementary material, Appendix B. Each of the neural networks we consider is implemented using PyTorch [31].

The learning methods considered in this work are each trained on one of the two target task formulations described in Section 3. For derivative-based prediction, the training is conducted supervised on ground truth snapshots gathered from the underlying models. For each system we randomly sample initial conditions and each of these is then numerically integrated to produce a trajectory. Each trajectory includes state samples $x$ as well as target derivatives $\dot{x}$ used for training. For direct prediction, we no longer require numerical integration; instead we directly predict the trajectory in a sequential fashion. In this setting, we approximate $\tilde{f}_\theta(x(t)) \approx x(t + \delta_t)$ for a discrete time step size $\delta_t$. For the derivative prediction task we report results using the leapfrog integrator. Full results using other numerical integration schemes are available in the supplementary materials.

We pick the same set of learning methods and apply it to both tasks independently to judge performance in each. For many systems the state is divided into position and momentum components: $x \equiv (q, p)$. For the Navier-Stokes problem, the state $x$ is made up of the flow velocity field, and the scalar field for pressure. After training, we produce rolled-out trajectories from held-out initial conditions, either by combining with a numerical integrator in the case of derivative prediction, or in a directly recurrent fashion in the case of step prediction. Each neural network is instantiated in three independent copies, each of which is trained and evaluated across all sampled trajectories. We compute a per-step MSE against a ground truth value, average these per-step MSEs to produce a per-trajectory error, and record these errors for analysis. Our experiments are designed to test several aspects of physical simulation. We highlight the most salient ones next, and report more extensive results in Appendix C.

**Training set size**   In general ML problems, one would expect additional training samples to systematically improve (in-distribution) evaluation performance. However, Figure 2 illustrates a clear

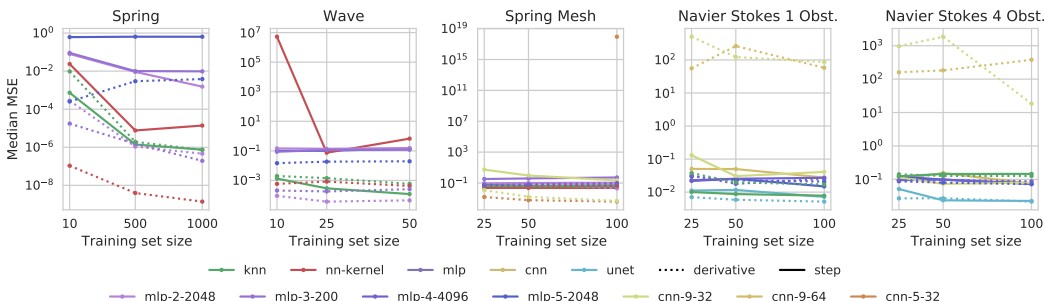

Figure 2: Median MSE error with respect to the training set size for each of our system configurations. We show varying architecture choices for each method.

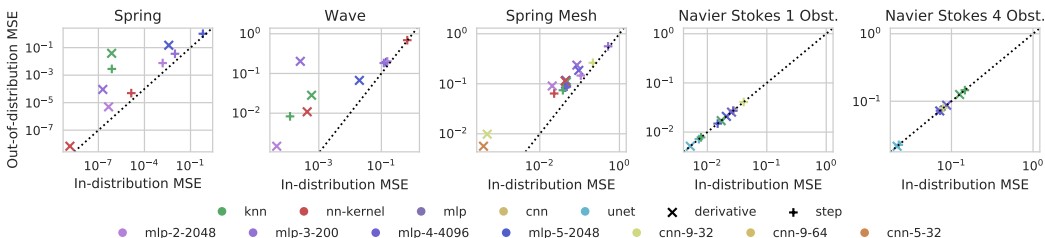

Figure 3: Median MSE for in-distribution evaluation sets vs. out-of-distribution evaluation sets for each system. Colors represent the same method and are varied for different architecture choices. Marker shapes distinguish step and derivative prediction, and the dotted line is the identity line. Outliers for the spring mesh and both Navier-Stokes configurations were removed. Values on both axes were approximately $10^{13}$ for the spring mesh and in the range $10^2$–$10^3$ for Navier-Stokes.

saturation of performance on the simplest systems when using neural networks as function approximators, in contrast with non-parametric KNNs and the kernel method. We attribute this saturation to an inherent gap between the training and evaluation objectives. While data-driven methods are optimized to minimize next-step predictions, the final evaluation requires built-in stability to prediction errors. Including regularisation strategies to incorporate stability, such as noise injection [34], is shown to help, but not fully resolve this issue.

**Out-of-distribution evaluation**   For simplicity, we only examine the out-of-distribution error for networks trained on the largest training set size. The added challenge of out-of-distribution samples varies with the construction of each system. It is possible to get some idea of the difficulty increase by examining the accuracy penalty for the KNNs, and comparing it to how well the more advanced models are able to generalize.

Benefits of neural networks for generalization over KNN are visible across several systems in Figure 3, particularly in the spring system for small MLPs for derivative prediction and nn-kernel in both cases. The KNN suffers a significant increase in error while these methods produce only somewhat worse predictions. Benefits are still present, though less pronounced, for the wave system derivative prediction where neural networks increase in error, but the kernel method and small MLP maintain a lower absolute error than the KNN. On the Navier-Stokes systems none of the methods suffers an increase in error for out-of-distribution evaluation. The change in radius distribution for the obstacles did not pose an additional challenge sufficient to produce a measurable change in error distribution. We attribute this to low dimension of the initial condition space.

**Step and derivative prediction**   The step and derivative prediction instances of each learning problem lead to different accuracy from the learning methods we test. While most physical systems are naturally described in terms of their derivatives through corresponding ODEs/PDEs, data-driven simulations also offer the alternative of bypassing this differential formulation and predict the next state directly. Such 'cavalier' approach avoids the compounding error amplification effects across

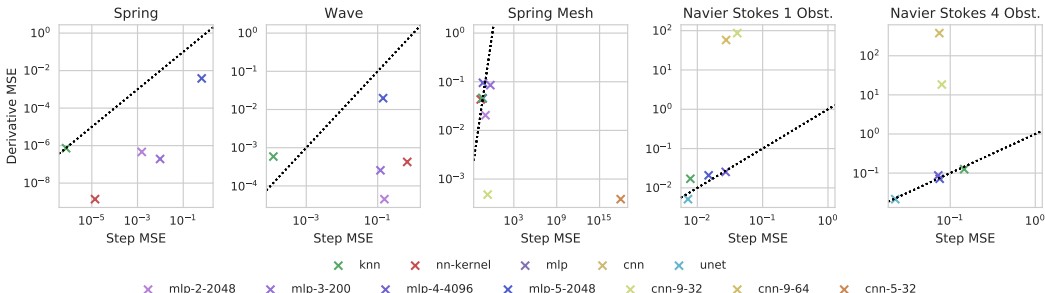

Figure 4: Median MSEs for derivative vs. step prediction on the same evaluation set. Results are displayed for each of our system configurations. The dotted line is the identity line.

integration steps, at the expense of sample efficiency. Figure 4 illustrates these tradeoffs across our systems.

An important example of this effect is the performance of CNNs on the spring mesh system (Figure 8 in the Appendix). When working through a numerical integrator and performing derivative prediction they produce the lowest error of all methods tested, but following the same training protocol for step prediction these architectures produce high errors, or are unstable. This case is likely an interaction of the architecture with the specific learning task. For the spring mesh, step prediction requires outputting the position of the particle which requires manipulating its global coordinates, while derivative predictions allow the network to more easily act locally and compute only a relative motion for the particle. The derivative prediction task better takes advantage of the spatial invariance of the CNNs. This difference in performance reflects the importance of tailoring architectures to the specific task, and some potential for neural network architectures to benefit from incorporating knowledge of a system's behavior.

**System and dataset complexity**    Several trends we observe correlate with the difficulty of learning to simulate a system, and the variation in its behavior across the training and evaluation samples. This is generally a combination of the system's state dimension, and variation in its behavior, approximated by the dimension of the distribution from which initial conditions are sampled.

This is particularly visible in Figure 5 in the performance of the KNN methods, and, in many cases, the performance of simpler methods such as the small MLPs. On the simpler systems, such as the spring and wave, the KNNs generally perform well because even though the wave system has a relatively large state dimension of 125, like the spring its initial condition is sampled from only two parameters and its behavior can be readily predicted from these. The Navier-Stokes system with a single obstacle is another instance of this sort of behavior: the KNN is readily able to reproduce flows it has not seen because a sampling of 100 obstacle positions is such that an evaluation sample is close to a trajectory seen at training time. Therefore, small MLPs and the kernel method produce similar performance. When the difficulty is increased by sampling four obstacles, the KNN and MLP performances suffer, and larger networks such as the u-net are needed to maintain approximately the same performance.

**Choice of numerical integrator**    For our derivative prediction tasks we combine our trained methods with three explicit integrators with orders 1, 2, and 4. In most of our systems these produce at most a small increase in accuracy, holding all other training and evaluation parameters equal. However on the Navier-Stokes system the higher order integrators produce somewhat higher errors, particularly for the u-net and the MLPs. This appears related to the approximated derivatives used for training this system. The learned derivatives produce some small deviations which are compounded when combining multiple derivative samples.

**Computational overheads**    Another important aspect to consider when applying learning methods to physical simulation problems is the time required to compute each step, and the computational overheads introduced by the lack of knowledge of the true system physics. With standard numerical integration methods, it is generally possible to improve the quality of generated trajectories by decreasing the size of the time step used during integration. We take advantage of this in order to

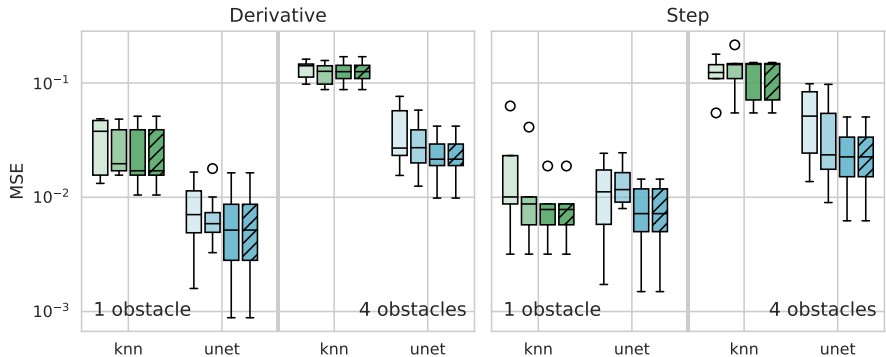

Figure 5: MSE distribution across trajectories in the evaluation set for KNNs and u-nets on our Navier-Stokes system, both the formulation with one obstacle, and the settings with four, as well as for derivative and step prediction. Box plot mid-lines show medians, box area is between first and third quantiles, whiskers extend $1.5\times$ beyond the boxes, outliers are plotted past the whiskers. Darker colors denote increasing training set size. The final hatched box is the same network from the final un-hatched box, tested on an out-of-distribution evaluation set.

Table 2: Time comparison for derivative prediction against baseline numerical integrators

| System | Architecture | Euler | | Leapfrog | | RK4 | |
|---|---|---|---|---|---|---|---|
| | | Time Ratio | Scaling | Time Ratio | Scaling | Time Ratio | Scaling |
| Spring | knn | 367.0 | 1× | 405.8 | 16× | 311.3 | 64× |
| | nn kernel | 180.7 | 1× | 198.6 | 1× | 173.3 | 1× |
| | mlp-2-2048 | 185.8 | 1× | 191.6 | 16× | 177.7 | 64× |
| | mlp-3-200 | 237.6 | 1× | 237.5 | 16× | 227.2 | 64× |
| | mlp-5-2048 | 473.8 | 4× | 369.6 | 64× | 360.9 | 128× |
| Wave | knn | 24,102.7 | 8× | 16,945.1 | 256× | 16,132.0 | 256× |
| | nn kernel | 35.3 | 8× | 22.3 | 256× | 19.9 | 256× |
| | mlp-2-2048 | 25.4 | 8× | 16.5 | 256× | 14.2 | 256× |
| | mlp-3-200 | 31.0 | 16× | 19.7 | 256× | 17.8 | 256× |
| | mlp-5-2048 | 60.5 | 16× | 38.0 | 256× | 34.6 | 256× |
| Spring Mesh | knn | 708.6 | 8× | 626.1 | 128× | 690.4 | 256× |
| | nn kernel | 5.3 | 8× | 4.4 | 128× | 4.8 | 256× |
| | mlp-2-2048 | 3.0 | 8× | 2.6 | 128× | 2.8 | 256× |
| | mlp-3-200 | 3.4 | 8× | 3.2 | 128× | 3.4 | 256× |
| | mlp-4-4096 | 7.8 | 16× | 7.1 | 128× | 7.7 | 256× |
| | mlp-5-2048 | 7.1 | 8× | 6.2 | 128× | 5.3 | 256× |
| | cnn-9-32 | 11.0 | 2× | 10.5 | 32× | 11.3 | 64× |
| | cnn-5-32 | 7.4 | 1× | 9.2 | 32× | 7.3 | 64× |

estimate the time overheads of our learning methods relative to our baseline numerical integrators at approximately corresponding error levels.

We numerically integrate each system at time step sizes scaled by powers of two. For each trajectory in the derivative prediction setting, we find the smallest scaling factor at which the numerical integrator exceeds the learning method's error at their final shared time step, approximating the factor by which numerical integration can be made faster until it begins to underperform the learned method.

Table 2 reports the results of these experiments. For each numerical integrator, the "scaling" column reports the most common scaling factor found for each trajectory. The "time ratio" column represents the learned method's evaluation overhead (median times, counting only per-step network evaluation costs, not numerical integration or data transfers). Note that the numerical integrator makes fewer steps than the learned method so the overall trajectory time must be further adjusted by the scaling factor.

In general, the neural networks are slower per-step by one or two orders of magnitude. KNNs are slower by significantly larger factors, particularly for the wave system. This is likely partially due

to the default scikit-learn KNN implementation used, and due to the large size of the wave system training sets (large state dimension and large number of training snapshots). Scaling factors increase with the order of the integrator as higher-order integrators are more tolerant of large step sizes and maintain low error.

It is likely that these overheads could be reduced with more optimized implementations of both the numerical integrators and learned methods. The derivative prediction task is also constrained by its need to interact with the numerical integrator. In this setting the learned methods cannot be expected to outperform the quality of the solutions generated by the true system derivatives. This reflects a penalty resulting from a lack of knowledge of the true underlying system, and a penalty for learning from observations in this case. Step prediction without involving the numerical integrator potentially avoids some of these constraints, if learning is successful.

# 6   Conclusions and limitations

The results in this work illustrate the performance achievable by applying common machine learning methods to the simulation problems in our proposed benchmark task. We envision three ways in which the results of this work might be used: **(1)** the datasets developed here can be used for training and evaluating new machine learning techniques in this area, **(2)** the simulation software can be used to generate new datasets from these systems of different sizes, different initial condition dimensionality and distribution; and the training software could be used to assist in conducting further experiments, and **(3)** some of the trends seen in our results may help inform the design of future machine learning tasks for simulation.

For the first and second groups of downstream users, we have made available the pre-generated datasets used in this work, as well as the software used to produce them and carry out our experiments. These components allow carrying out the measurements we have made here, and permit further adjustments to be made. Documentation on using the datasets and the software is included in Appendix D.

For the third group, we highlighted a few trends that suggest useful steps to take in developing new problems and datasets in this area. First, we advise including several simple baseline methods when designing new tasks. In particular the inclusion of standard numerical integrators (for derivative-type problems) and KNNs are useful to evaluate the difficulty of the proposed task. Specifically, KNNs are useful for examining the performance achievable by memorizing the training set, and are thus *witnessing* an appropriate design of data distribution that captures the true high-dimensionality of the prediction task. As an example, in the Navier-Stokes examples some task formulations may inadvertently be simple to memorize, even if the complexity of the system itself may not immediately suggest it. The numerical integrators are likewise useful as baselines both to ensure that the derivative learning is feasible even when achieving no error in predictions, and also to evaluate the penalty in accuracy which is incurred by operating without access to the true physics. We believe that in light of these observed trends, including baseline methods such as standard numerical integration schemes and simple learning methods such as the KNN is important in understanding tasks in this area. Including these assists in experiment design by helping to calibrate the difficulty of a target task.

**Limitations**   While our benchmark provides actionable conclusions on a wide array of simulation domains, it is currently focused on temporal integration, and as such it does not cover important settings in scientific computing. For instance, we do not currently include an instance of a surrogate model, which could provide different tradeoffs benefiting ML models. Additionally, we have focused on two setups for data-driven simulation (differential snapshot prediction and direct snapshot prediction), but other alternatives exist that might mitigate some of the shortcomings we observed; for instance by considering larger temporal contexts (as in [4]), as well as enforcing certain conservation laws into the model [8, 4]. Finally, while we report some measurements of timings and relative computational overheads, there are other dimensions to the time-accuracy tradeoff which remain to be explored and further software optimizations are most likely possible.

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
