# OpenReview forum: "An Extensible Benchmark Suite for Learning to Simulate Physical Systems"
_NeurIPS.cc/2021/Track/Datasets_and_Benchmarks/Round1 — NeurIPS 2021 Datasets and Benchmarks Track (Round 1)_

### Official Review · Reviewer_PRGA · 2021-07-03
**Interesting work, but the value of data and benchmarks is unclear.**

**Rating:** 7
**Confidence:** 3
**Correctness:** Yes.
**Clarity:** Yes.

**Strengths:**

- This is hard work (850h on GPU + 2180h on CPU). It is really difficult to process such a large amount of data and keep it unified and correct.

- The data, source codes, and documents are all well organized.

- The generation methods, documents, and experiments are described in detail.

**Weaknesses:**

For a paper of dataset and benchmark, the contributions should be novel/originality data, comparison and analysis of state-of-the-art methods, and the impact on the future of related fields. I hope the authors could give more explanation about these points.

- The paper claimed that the proposed benchmark takes a step towards unified evaluation protocols and metrics, but it is unclear.

- Data. The complex system is more important for simulation problems. The value of simple system data needs to be explained. The data of four systems are also generated in the existing paper, such as spring in [7], Navier-Stokes in [48]. What is the difference between the provided data and the previous data?

- Benchmarks. Only simple baseline models are tested (except for u-net on naiver-stokes), and the paper claim that these models could not solve the problem well. However, advanced models have been proposed and compared with the baseline models in the existing papers. SOTA models should be tested and analyzed.

**Additional Feedback:**

- more references should be cited in the Introduction to clarify the current problem.
- difference with the previous works should be described.


**Documentation:**

Yes.

**Ethics:**

No.

**Relation To Prior Work:**

No.

**Summary And Contributions:**

This paper introduced a benchmark for the methods of data-driven-based physical system simulation. They provided a dataset of four physical systems (spring, wave, spring mesh, navier-stokes) including data, data generation methods, codes, and documents. Then, they conducted a variety of experiments using knn, nn kernel, MLPs, and CNNs. This work is interesting, and important for learning physical systems. The descriptions are also detailed. However, the value of this benchmark needs to be more explained, please see Weaknesses.

---

> ### Author Response · Authors · 2021-07-11
> **Reply to reviewer comments (part 1/2)**
>
> Thank you for your thoughtful review.
>
> > The paper claimed that the proposed benchmark takes a step towards unified evaluation protocols and metrics, but it is unclear.
>
> In this work we propose a set of methods which we use across all our tests. The test problems we propose span elliptic and hyperbolic systems with a wide range of complexity, as measured by their state dimensions and the distribution of sampled initial conditions. Despite this variety we keep a consistent test protocol, several parts of which we find particularly valuable and highlight here:
>
> - We keep a similar training and evaluation structure across each of our test systems and across two learning task formulations (learning a right hand side function to predict time derivatives, and learning a time-stepping scheme directly in our step prediction setting). These are each trained on the same sampled data, and evaluated analogously with as few differences as possible.
>
> - In our evaluations we also include several fundamental/traditional approaches as baselines. In particular, the KNNs which help to measure the difficulty of the proposed task (and indeed did help with this on the single-obstacle Navier-Stokes setting, prompting us to add a multi-obstacle version), as well as the standard time integrators which offer comparisons against typical methods and help gauge the gap in solution quality relative to integrating the system with a standard non-learned approach.
>
> - The tests we perform with specified boundary/initial conditions make it possible to compare results across different learning methods. We have also made these particular initial conditions and snapshots available for use by others to permit close comparisons against the results we observe here, along with the code used to generate the data and perform our experiments to permit others to modify these settings or recreate the results following our procedures. This would make it possible for other works to run comparisons against these results and permit simpler comparisons across various results.
>
> While the quest for unification can of course be pushed even further, we believe our methodology is already providing value, e.g. because it allows us to calibrate the relative difficulty of a physical simulation task. We will redraft our discussion in the paper to make sure this comes across more clearly.
>
> > The value of simple system data needs to be explained.
>
> We fully agree with the reviewer that complex systems are those with more practical/real-world applications and relevance. We chose to compliment these complex systems in our benchmark set with those across a wider range of complexity. The simpler systems can serve as unified tests to assess the relative value of certain ML architectures and to study their stability to e.g. distribution shifts and large timescales, and the impacts of state dimensions vs. complexity of a system's initial distribution on network performance. We believe that this initial assessment of these questions for simple systems is an important prerequisite to properly study the behavior of data-driven methods on complex systems.
>
> > What is the difference between the provided data and the previous data?
>
> The spring system has been used in similar forms in past works. It is a standard benchmark and we included it for completeness. Our Navier-Stokes problem includes a multi-obstacle formulation which we have not seen in other works. This was added after our tests with the KNN suggested that the single-obstacle version may be unexpectedly straightforward for the KNN. So this setting added in light of those results is new.
>
> For all systems, we also include out-of-distribution samples for evaluation which is another aspect of our suggested protocol that we have not seen in previous work. We also make available all snapshots and the software used to produce them for use and modification by others. Other works do not always universally make available all of their data snapshots or all software used to produce them, and we have made a special effort to ensure that our experimental settings can be used exactly as we have, or modified/re-sampled and used as a base for future work.

---

> ### Author Response · Authors · 2021-07-11
> **Reply to reviewer comments (part 2/2)**
>
> > Only simple baseline models are tested (except for u-net on naiver-stokes) ... SOTA models should be tested and analyzed.
>
> The reviewer is completely right that there are many other models that can be applied to and tested with our proposed benchmark problems, and this has high priority for us as something to pursue using our framework here. In the current work, we focused on providing numerical results that give insights into how to design benchmark problems and that provide baselines for future comparisons. As far as we are aware, many of the state of the art methods are significantly tailored to a particular task formulation and are less generally-applicable. Several recent state of the art methods do not all have full source code posted. In particular we worked to reimplement a graph network-style architecture for testing, but we were unfortunately unable to reproduce the performance, and had difficulty resolving this without source code which the authors were unable to release due to internal review requirements.
>
> We believe that this reinforces the need for an open benchmark, that we hope will facilitate other researchers (who may not have the freedom to share their code) to test their model against our well-calibrated baselines and permit more straightforward comparisons of results.
>
> > more references should be cited in the Introduction to clarify the current problem.
>
> We will include additional references and discussion in our introduction to make clearer the variety of problems used in other works, and some of the resulting challenges involved in comparing results in this area.
>
> We hope that the above helps clarify. We're happy to redraft contents of the paper to improve clarity there as well, and of course, please let us know if you have any other questions.

---

> > ### Comment · Reviewer_PRGA · 2021-07-14
> > **Thank you for the explanation**
> >
> > Thank you for the explanation. I will change my rating.

---

> > > ### Comment · Reviewer_PRGA · 2021-07-17
> > > **I have changed my rating from 5 to 7.**
> > >
> > > I have changed my rating from 5 to 7.

---

### Official Review · Reviewer_4TfX · 2021-07-05
**An attempt to provide unified benchmarks for neural net / ml models in scientific computation involving physics based model simulations.**

**Rating:** 7
**Confidence:** 4
**Clarity:** The paper is very well written and cl…

**Strengths:**

-- Provides a standard benchmark for contrasting traditional solutions for numerical computation with physics based models against data-driven methods.
-- I like the progression of complexity of the benchmarks and the exploration of narrow and wide data regimes. Such a standard benchmark allows for systematic evaluation and tradeoff analysis.
-- The experimental results provide sufficient insights
-- The code and data is made accessible and is extensible
-- No ethical/social implications exist

**Weaknesses:**

-- Apart from highlighting the deviations in results between data driven and traditional methods, It is not absolutely clear how the benchmark results points to ML algorithm improvements.  (The authors state this as a point of strength of the paper, but it is not obvious what they imply by this).
-- While I like the organization of the benchmark in the paper, I would have assumed that a more simpler differential system with well known closed form solution for the result may be an interesting option as a baseline.
-- The paper will be strengthened significantly if other aspects not covered by the authors (described in the limitation sections) are included including the missing timing analysis.

**Additional Feedback:**

Can you comment about the points in the weakness section?

**Correctness:**

The claims appear to be correct and the experiments are conducted in a sound way. The experimental design and evaluation is fine.

**Documentation:**

There is sufficient detail with respect to code, data generation, availability, and extensibility.  There is sufficient detail to reproduce the experiments.

**Ethics:**

none.

**Relation To Prior Work:**

The references to prior work is adequate.   I have seen a number of papers in the field with each of them doing their own evaluation and the current paper is attempting to have a coherent benchmark covering a range of problems.  I am absolutely not sure if there are other relevant benchmark papers in the field that are not cited.

**Summary And Contributions:**

The present work is motivated by the need for a) thorough evaluation of data-driven approaches in scientific computing pipelines and b) the lack of standardized benchmarks in the literature.   The authors focus on physical simulation benchmarks that a) map a high-dimensional state space into another high-dimensional space (as in temporal integration schemes, mapping the state of the system at one time step to the next, or b) from a high-dimensional input space to a lower-dimensional output (as in surrogate models, mapping the initial conditions to a functional of the solution.  In addition the paper addresses the narrow data regime, where initial conditions are sampled from a low-dimensional manifold (even within a high-dimensional state space), and the wide regime, where initial conditions span a truly high-dimensional space.

The main contribution of the paper is the presentation of a suite of simple, representative physics problems along with reference numerical solutions for traditional time integration schemes to benchmark data-driven methods (MLPS, CNNs, kernel machines, Nearest neighbors).  The key conclusion of the paper is that, even in the simplest physical models, current
data-driven pipelines, while providing qualitatively acceptable solutions, are quantitatively far from
directly numerically integrating physical models, and this performance gap appears unfeasible to
close by merely scaling up the models and/or the dataset size.   Another finding is that a simple L2-based nearest neighbor regressor outperforms most deep learning models in the narrow regime for complex systems such as incompressible Navier-Stokes systems.

---

> ### Author Response · Authors · 2021-07-11
> **Reply to reviewer comments**
>
> Thank you for your thorough review.
>
> > It is not absolutely clear how the benchmark results points to ML algorithm improvements.
>
> As far as the usefulness of the trends we see for improving ML methods, we mean in particular some considerations for designing or selecting target tasks in light of the results we illustrate in this work. For example, the success of KNNs and other simple methods may illustrate that it is easy to inadvertently design a problem which is surprisingly low-dimensional and easier for a KNN to handle, even if the task itself appears challenging (for example the single- vs. multi-obstacle Navier-Stokes settings). Based on our experience with these tests we believe including some of these simple baselines is important to understand and calibrate the difficulty of the target task. We will redraft our concluding comments to make sure that we're more clear about that.
>
> > more simpler differential system with well known closed form solution for the result may be an interesting option as a baseline
>
> The spring system in our set of problems is of this sort even though we do generate snapshots with a numerical integrator. We configured our test systems to provide high-quality ground truth data even if it is produced with a numerical integrator. When generating with these, we calibrated our time step sizes by examining the convergence of the snapshots as the step sizes were decreased so that we could be confident that our training and reference snapshots were of a high quality. Even though these snapshots (particularly for the spring system) are not produced from closed form solutions, we worked to ensure that they are still high quality snapshots. We will include a description of the spring system closed form in the paper for completeness and for reference by users of the benchmark suite.
>
> > other aspects not covered by the authors (described in the limitation sections) are included including the missing timing analysis.
>
> We will finish up our timing analysis of the computational overheads of the data-driven methods vs. baseline integrators and add it to the work.
>
> > I am absolutely not sure if there are other relevant benchmark papers in the field that are not cited.
>
> We aren't aware of prior works which try to assemble a general set of benchmark tasks. In our experience as well, most works---naturally enough---choose tasks to suit their proposed method or architecture. Our hope with this work is to provide a general set of tasks which are widely useful, as well as some trends we observed in running our tests which may be useful even in cases which do not use the set of tasks we propose here.

---

> > ### Comment · Reviewer_4TfX · 2021-07-13
> > **The clarification is fine and confirms my rating of Good paper, Accept.**
> >
> > Thank you for the point by point clarification.  It confirms my original rating.

---

### Official Review · Reviewer_YUCr · 2021-07-06
**Concise and Clear Benchmark for Physical Simulations**

**Rating:** 8
**Confidence:** 4
**Clarity:** The paper is very well written and cl…

**Strengths:**

The paper is very concise, easy-to-follow and well-illustrated.

The authors do a great job motivating the four representative benchmark physical systems, and provide a comprehensive array of baseline data-driven methods. While neither of these are exhaustive, the flexibility of their framework allows for the integration of other learning tasks or machine learning methods.

All in all, their contribution promisingly lays the groundwork for future research in the field of scientific computing

**Weaknesses:**

While the graphs are typically easy-to-follow,  I was slightly confused by repeated colors in figures 3 and 4. My understanding is that same-color marks are different architectures of the baseline methods. However, different architectures could have significantly different strengths and weaknesses. For instance, shallow MLPs are typically more robust to noisy datasets than deep MLPs.

Do you believe that readers would benefit from having more fine-grained labels for the methods (e.g. shallow vs deep MLP/CNN) in figures 3 and 4?

Further, due to the non-deterministic nature of some NN-based approaches, it would make sense to average results over multiple runs. In the main paper there doesn't seem to be any indication of this. Were the results presented as averages over multiple runs?

**Additional Feedback:**

MSE seems like a reasonable metric to evaluate the trajectories of simulations. However, my understanding is that predictions close to $t=0$ are more likely to be correct, and that errors compound by letting the system run for longer.  If this is correct, a weighted squared error (e.g. decaying squared-error) could be a more suitable metric for longer trajectories. However, I understand that this would've added an extra dimension of complexity to the overall paper.

**Correctness:**

The 4 benchmark systems are motivated and thoroughly detailed. The increasing level of complexity of the different datasets seems like a reasonable choice for the project.

The data-driven methods discussed represent standard approaches in the field of physical simulations.

**Documentation:**

Data collection is based on physical simulations, which obtained using a variety of computational tools such as NumPy and SciPy. This is very clearly detailed. Reproducibility seems straightforward.



**Ethics:**

No ethical concerns.

**Relation To Prior Work:**

While previous work is discussed in section 2, there is no explicit description of any physical system benchmark. Is this because because all prior work has had a major lack of standardization? One or two examples of how prior work has been more unstructured could strengthen your motivation.

**Summary And Contributions:**

The authors introduce a physical systems evaluation dataset framework that focuses on evaluating machine learning algorithms for simulation problems. They provide four systems of increasing difficulty to evaluate baseline methods (spring, wave, Naive-Strokes and spring mesh), explore the trade-offs between derivative-based prediction and step prediction and advocate the usage of K-nearest-neighbors to better understand the complexity of different simulation tasks.

---

> ### Author Response · Authors · 2021-07-11
> **Reply to reviewer comments**
>
> Thank you for your detailed review.
>
> > Do you believe that readers would benefit from having more fine-grained labels for the methods (e.g. shallow vs deep MLP/CNN) in figures 3 and 4?
>
> Yes, we agree. Just for clarity, the figures in the main body of the paper are all subsets of the results presented in the box plots in the appendices. We will split out the different architectures in the scatter plots in the main paper to make it possible to separate the different architectures within the broader categories.
>
> > Were the results presented as averages over multiple runs?
>
> For this submission we had some constraints on available compute capacity when training our large sets of neural networks across our problems and architectures. The results presented here required several hundred hours of GPU training time and well over a thousand hours of CPU time, which limited our ability to repeat training during these experiments. The results in the paper are computed for one neural network training for each problem formulation (step vs. derivative prediction) for each system and architecture. The averages and distributions in the box plots are computed across the multiple sampled evaluation trajectories.
>
> We will start new training passes for the architectures presented here and incorporate those extended results.
>
> > Is this because because all prior work has had a major lack of standardization? One or two examples of how prior work has been more unstructured could strengthen your motivation.
>
> We aren't aware of other prior works which focus on producing sample tasks for general use. We will include some additional references to the types and variety of systems used in previous works.
>
> Some of the tasks we chose are related to sample problems chosen in other works (in particular the spring and single obstacle Navier-Stokes systems), and we also include some others that we haven't seen used as widely (the wave, spring mesh, and multi-obstacle Navier-Stokes).
>
> > ... predictions close to $t=0$ are more likely to be correct, and that errors compound by letting the system run for longer... a weighted squared error (e.g. decaying squared-error) could be a more suitable metric for longer trajectories
>
> Yes, that's correct. We do observe that errors tend to increase across the time steps as the system is integrated away from $t=0$. We tested other measurements but found that each had some drawbacks, and settled on MSE as a widely-used and simple measure. We will recompute some of our error distributions from our stored evaluation results and report these using another measure, as additional extended discussion.

---

> > ### Comment · Reviewer_YUCr · 2021-07-14
> > **The Explanation Reassures me that this is a Good Paper**
> >
> > Thank you for replying to my questions.
> >
> > Given the extensive training time required for the experiments, the computing constraints are understandable and make your benchmark all the more compelling. In the absence of averaged results, it may be beneficial to briefly mention any potential sources of variance in the results (e.g. which simulations you'd expect to have higher variance), just as a way giving more context to readers.
> >
> > I will be updating my score to 8: Top 50%

---

### Author Response · Authors · 2021-07-11
**Thanks to the reviewers**

Thank you to all the reviewers for your very thoughtful comments. We appreciate the time you spent reviewing our work. We have posted comments as separate replies to each of your reviews.

---

### Author Response · Authors · 2021-07-15
**Updated paper with review comments**

Thank you again to all the reviewers, we appreciate your comments. We've posted an updated copy of the main paper and supplementary materials in which we've worked to address comments from the review period: rewording and clarification as well as additional averaging in experiment runs and timing results.

---

### Decision · Program_Chairs · 2021-07-26

**Decision:**

Accept

**Comment:**

This paper introduces a physical evaluation dataset framework for scientific computing pipelines that map one high-dim state space into another high- or low-dim one, providing a suite of simple representative physics problems.  Reviewers appreciated for motivation, clarity, comprehensiveness, and overall contribution to the space.